

# No significant association of repeated messages with changes in health compliance in the COVID-19 pandemic: a registered report on the extended parallel process model

Jingwen Yang[1], Xue Wu[1], Kyoshiro Sasaki[2] and Yuki Yamada[3]

[1] Graduate School of Human-Environment Studies, Kyushu University, Fukuoka, Fukuoka, Japan
[2] Faculty of Informatics, Kansai University, Takatsuki, Osaka, Japan
[3] Faculty of Arts and Science, Kyushu University, Fukuoka, Fukuoka, Japan

## ABSTRACT

When people are confronted with health proposals during the coronavirus disease 2019 (COVID-19) pandemic, it has been suggested that fear of COVID-19 can serve protective functions and ensure public health compliance. However, health proposal repetition and its perceived efficacy also influence the behavior intention toward the proposal, which has not yet been confirmed in the COVID-19 context. The present study examined whether the extended parallel process model (EPPM) could be generalized to a naturalistic context like the COVID-19 pandemic. Additionally, we explored how repetition of a health proposal is involved with the EPPM. In this study, two groups of participants were exposed to the same health proposal related to COVID-19, where one group was exposed once and another group twice. Participants then filled out a questionnaire consisting of items concerning behavior intention and adapted from the Risk Behavior Diagnosis Scale. Structural equation modeling was used to determine the multivariate associations between the variables. Although the results showed that behavior intention is predicted by perceived efficacy, no significant influence of perceived threat was detected. Furthermore, no significant effect of repetition was found toward either response efficacy or perceived susceptibility. These findings indicate that to promote health compliance during the COVID-19 pandemic, it is more efficient to focus on health proposals' perceived efficacy rather than the disease's perceived threat. For future health communication research, the present study suggests improved analysis strategies and repeated manipulation of messages.

## INTRODUCTION

One year has passed since the COVID-19 outbreak, but the health threat has not been terminated yet. As of February 2, 2021, there are over 100 million confirmed cases globally; in Japan, 389,518 cases and 5,722 deaths (mortality rate: 1.47%) have been reported

Corresponding author
Jingwen Yang,
yangjingwen0921@gmail.com

(*World Health Organization, 2021*). During this period, while a heavy loss of both life and economy is caused, some countries and regions have achieved staged success in the fight against this disease. There is no doubt that public compliance with effective health proposals plays a crucial role in achieving this success.

It is suggested that functional fear, which is explained as one of the negative emotions serving protective functions in certain contexts, has promoted public health compliance during the COVID-19 pandemic (*Harper et al., 2020*). Nevertheless, this needs to be explored further, especially considering previous studies on health communication and the features of information dissemination in real life. The main question being addressed in our study is to examine how people's health compliance intention is influenced by various factors in the COVID-19 context in an exhaustive way.

It is known that fear has an impact on health compliance. In fact, fear-arousing communication is considered an effective way to promote health campaigns and has been widely investigated for promoting health awareness related to various topics, including smoking (*Leventhal & Watts, 1966*), alcohol use (*Wolburg, 2001*; *Moscato et al., 2001*), AIDS (*Treise & Weigold, 2001*) and so forth. After several initial studies, inconsistent results indicated that a simple monotonic function of fear might not be expected in persuasive health communication. Specifically, despite a large number of studies indicating the positive main effect of fear on persuasion, which means that fear improves persuasiveness (*Leventhal & Watts, 1966*; *Dabbs & Leventhal, 1966*; *Leventhal, Singer & Jones, 1965*), relatively few report the negative main effect of fear, showing as less attitude change with stronger threat (*Janis & Feshbach, 1953*; *Janis & Terwilliger, 1962*). Later research confirmed that the effects of fear-arousing communication interact with various source variables, message variables, and receiver variables and thus cannot be described easily (*Miller & Hewgill, 1966*).

One of the most recent and prevalent theories on fear-arousing communication is the extended parallel process model (EPPM: *Witte, 1992*, *1994*), which is based on former frameworks, including the Parallel Response Model (*Leventhal, 1970*) and protection motivation theory (*Rogers, 1975*). In the EPPM, there are four main factors, which influence the prediction of certain communication outcomes: perceived susceptibility and severity composing perceived threat, self-efficacy and response efficacy composing perceived efficacy. Perceived susceptibility refers to a belief about the risk of experiencing a threat, whereas severity refers to a belief about the magnitude of the threat. On the other hand, self-efficacy is defined as a belief about the ability to perform a recommended proposal to avert the threat; response efficacy is a belief about the effectiveness of the recommended proposal in deterring the threat (*Witte, 1996*). Concretely, when both perceived threat and perceived efficacy are high, people are most likely to engage in a danger control process, which means conforming to the recommended health proposal. Nevertheless, when perceived threat and perceived efficacy are high and low, respectively, people turn to a fear control process leading to coping responses that reduce fear and danger control responses. Meta-analyses on the results of fear-arousing research confirm the validity of the EPPM (*Floyd, Prentice-Dunn & Rogers, 2000*; *Peters, Ruiter & Kok, 2013*).
Currently, we are exposed to the same health proposals concerning COVID-19, in all likelihood, at least once in our daily lives, which makes message repetition an essential factor to be taken into consideration. Although research concerning stimulus repetition first emerged in the 1960s (*Zajonc, 1968*), there was no study on message repetition in persuasive communication until 10 years later. It is suggested that under moderate repetition (less than three times), agreement toward persuasive messages rises (*Cacioppo & Petty, 1979*). The rationale is explained as follows: Scrutiny is reinforced through a moderate level of message repetition, which enhances the understanding of message content and the merits advocated by the message, consequently improving supportive attitudes toward the message. Later research revealed that the mechanism mentioned above is applicable only when arguments in the message are perceived as strong and when the issue is of high personal relevance (*Cacioppo & Petty, 1989*; *Claypool et al., 2004*).

In health communication on COVID-19, we assumed that the interpretation of the content of a certain health proposal message might be related to factors in the EPPM. Concretely, response efficacy and perceived susceptibility might be perceived when reading a health proposal message. If the content of a health proposal was supported, high response efficacy should be found, resulting in a decrease in perceived susceptibility. Given the connection between the EPPM and message repetition in persuasive health communication and the actual state we have been through when confronted with health proposals during the COVID-19 pandemic, we built an integrative model ([Fig. 1]) to investigate the factors and their associations concerning an individual's behavior intention to conduct effective health proposals to prevent the infection.

The present study is of comprehensive significance in revealing the mechanism in the purview of public compliance with effective health proposals in the COVID-19 pandemic. It is unique and necessary in several aspects.

First, considering that COVID-19 is a real-life and ongoing public health emergency, some of its properties are fixed and thus cannot be manipulated. Take perceived threat, for example: In previous research, perceived threat was always considered together with fear because without extra fear-arousing materials (e.g., explanation of a certain disease in text or video), the perceived threat might not be notable enough to act as an independent variable. Unlike other unfamiliar diseases (e.g., melanoma), even though the degree to which one is influenced by the pandemic may vary, COVID-19 should be one of the immediate health threats in several countries and regions, including Japan, where the present study was conducted. Considering that fear-arousing information is no longer needed in the message, we are interested in exploring how perceived threat alone for COVID-19 affects behavior intention when there is no intentional manipulation of fear arousal (i.e., Hypothesis 8 below).

Second, there is still no research to test a model that combines the EPPM with message repetition. Although studies on similar topics have been conducted (*Skilbeck, Tulips & Ley, 1977*; *Treise & Weigold, 2001*; *Shi & Smith, 2016*), the results have not been analyzed in an integrated way or using structural equation modeling (SEM). To be specific, when supportive arguments toward the content of a recommended proposal are enhanced due to moderate repetition, it can also be interpreted as the change in response efficacy and
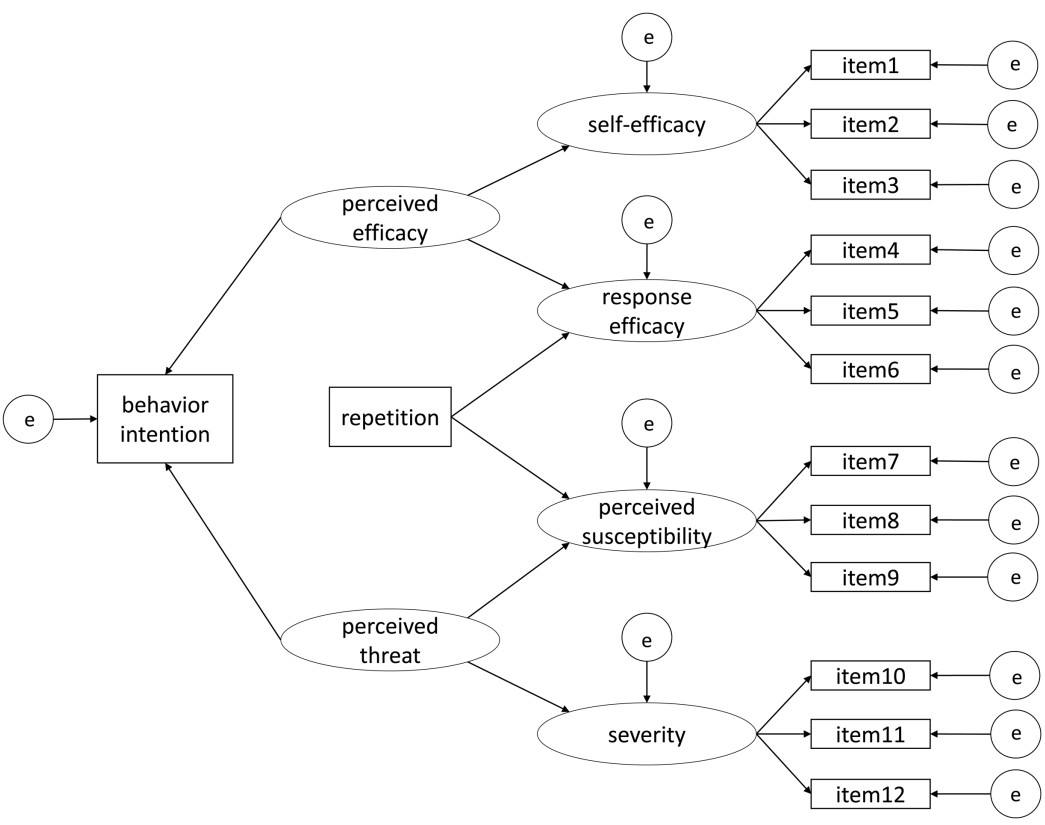

**Figure 1 An integrated model of health compliance intention in the COVID-19 pandemic.**

perceived susceptibility in the EPPM. Nevertheless, this connection between the two research topics has rarely been made clear, causing difficulty in making an exact prediction of the results.

Third, it remains essential to confirm the validity of EPPM's established construction, namely the sub-dimensions of perceived efficacy and perceived threat, to enhance compliance with public health guidelines in the COVID-19 pandemic. We assume better health compliance is due to higher perceived efficacy, but what does perceived efficacy mean? By learning more about its two indicators, self-efficacy and response efficacy, we can better understand why people choose to conform or not to certain health proposals. For instance, even though physical distancing is considered efficient in preventing infection (high response efficacy), the difficulty in keeping physical distance may vary between people based on their socializing needs (divergent self-efficacy), which results in different levels of health compliance. Similarly, if we are aware that perceived threat is indicated by perceived susceptibility and severity, we may know which properties to emphasize in education on COVID-19 to boost public health compliance.

Our hypotheses are as follows.

1) Hypothesis 1: Perceived efficacy has a positive effect on self-efficacy.
H0: Perceived efficacy has no effect on self-efficacy.

2) Hypothesis 2: Perceived efficacy has a positive effect on response efficacy.
H0: Perceived efficacy has no effect on response efficacy.

3) Hypothesis 3: Perceived threat has a positive effect on perceived susceptibility.
H0: Perceived threat has no effect on perceived susceptibility.

4) Hypothesis 4: Perceived threat has a positive effect on severity.
H0: Perceived threat has no effect on severity.

5) Hypothesis 5: Repetition has a positive effect on response efficacy.
H0: Repetition has no effect on response efficacy.

6) Hypothesis 6a: Repetition has a positive effect on perceived susceptibility.
H6b: Repetition has a negative effect on perceived susceptibility.
H0: Repetition has no effect on perceived susceptibility.

7) Hypothesis 7: Perceived efficacy has a positive effect on behavior intention.
H0: Perceived efficacy has no effect on behavior intention.

8) Hypothesis 8a: Perceived threat has a positive effect on behavior intention.
H8b: Perceived threat has a negative effect on behavior intention.
H0: Perceived threat has no effect on behavior intention.

## MATERIALS & METHODS

The present study received approval from the psychological research ethics committee of the Faculty of Human-Environment Studies at Kyushu University, Fukuoka, Kyushu, Japan (approval number: 2019-034).

### Preregistration

The research protocol of this study was peer-reviewed and registered prior to data collection at https://peerj.com/articles/10318/.

### Participants

The required sample size of $N = 301$ was set based on our preliminary experiment, using the findRMSEAsamplesize function in R (*MacCallum, Browne & Sugawara, 1996*), with $\alpha = 0.05$, power = 0.95, rmsea0 = 0.05, rmseaA = 0.01, $df = 70$. The degrees of freedom was calculated using the following formula: $df = (1/2) \{p (p + 1)\} - q$ (*Weston & Gore, 2006*), where $p$ is the number of observed variables (i.e., 14 in the model in Fig. 1), and $q$ is the number of parameters to be estimated (i.e., 35 in the model). In the present study, we recruited 521 participants via Yahoo! Crowdsourcing (http://crowdsourcing.yahoo.co.jp/). Since it is impossible to obtain written consent in anonymous online surveys, we explained the details of the online experiments and ethical matters in the instructions. The participants were asked to take part in the

experiments only when they agreed to instructions. We excluded 195 participants' data based on the following pre-defined data exclusion criteria:

1. To identify distracted respondents or satisficers (*Chandler, Mueller & Paolacci, 2014*; *Oppenheimer, Meyvis & Davidenko, 2009*; *Sasaki & Yamada, 2019*), we inserted a simple question as an attention check question (ACQ) in the middle of the questionnaire in the second wave. We did not add an ACQ in the first wave because the first wave was conducted only to control the frequency of exposure to the message, and thus the data in the first wave was not analyzed. The ACQ in the second wave was: Please choose number "2" from below. Consistent with other items, the ACQ also used a 7-point scale. The data from participants who chose 1 or 3–7 was excluded.

2. To ensure that our manipulation on the frequency of exposure is valid, the data from those who had seen this proposal before was excluded. We asked all participants the following question: "Have you seen this message before?" right after they were exposed to the message for the first time; we asked the question in the first wave of the repetition condition and in the second wave of the no-repetition condition. The data from participants whose answer was "Yes" was excluded.

3. For participants in the first wave of the repetition condition, a multiple-choice question concerning the content of the message ("What the message is about") was asked to confirm whether participants had read the message carefully enough to capture its meaning. Data from participants who gave a false answer was excluded.

4. Before analyzing the data, we calculated the standard deviation (*SD*) of each participant's scores on the 13 items in the second wave and excluded data from participants whose *SD* was zero. We performed this data exclusion because an *SD* equal to zero meant the same score on different items measuring divergent properties, which was strange and not possible.

5. Data of participants whose nationality was not Japanese and whose age was under 18 was excluded based on their answers to the relevant questions.

6. There was an open-ended question on the experiment's real purpose at the end of the questionnaire under both conditions in the second wave. The question was optional, and the data from participants who gave correct answers (consistent with the experiment's real purpose) was excluded.

After excluding those who met the exclusion criteria, 326 participants' data was included in the data analysis (141 females, 180 males, 5 others, $M_{age}$ = 45.8). Subsequently, participants were allocated to the no-repetition ($N$ = 166) and repetition conditions ($N$ = 160). Despite not matching the exclusion criteria, we found one person's data suspicious due to the answers provided (i.e., "99 years old," "other gender," and "one or two" for all questions in the second wave). Based on the registered data exclusion criteria and the fact that the inclusion of the person's data did not influence the results, we did not exclude it from our analyses.

**Table 1 Items included in the questionnaire (items 1–12 based on the risk behavior diagnosis scale, item 13 on behavior intention).**

*Self-efficacy*
  (1-strongly disagree, 7-strongly agree)

1   I am able to perform the underlined proposal to prevent the infection of COVID-19

2   It is easy to perform the underlined proposal to prevent the infection of COVID-19

3   I can perform the underlined proposal to prevent the infection of COVID-19

*Response efficacy*
  (1-strongly disagree, 7-strongly agree)

4   Performing the underlined proposal prevents the infection of COVID-19

5   Performing the underlined proposal works in deterring COVID-19

6   Performing the underlined proposal is effective in getting rid of COVID-19

*Perceived susceptibility*
  (1-strongly disagree, 7-strongly agree)

7   I am at risk of being infected with COVID-19

8   It is possible that I will get infected with COVID-19

9   I am susceptible to COVID-19 infection

*Severity*
  (1-strongly disagree, 7-strongly agree)

10  COVID-19 is a serious threat

11  COVID-19 is harmful

12  COVID-19 is a severe threat

*Behavior intention*
  (1-strongly disagree, 7-strongly agree)

13  In the future, when sanitizing my hands with alcohol-based hand sanitizer, I will press the pump slowly to the bottom to get a sufficient amount

## Measures

### Dummy questionnaire

We used the Need for Cognition Scale (*Kouyama & Fujihara, 1991*) as the dummy questionnaire in the first wave of the survey for both the repetition and no-repetition conditions. We did not analyze the data from the dummy questionnaire because it was irrelevant to our hypotheses.

### Risk behavior diagnosis scale

We adapted twelve items from the Risk Behavior Diagnosis Scale (*Witte, 1996*) to fit the target health proposal message in the present research and the context of COVID-19 (Table 1, items 1–12). The scores on the items were primary dependent variables. Concretely, self-efficacy and response efficacy toward the health proposal message were measured by items 1–3 and 4–6, respectively; perceived susceptibility and severity concerning COVID-19 were measured by items 7–9 and 10–12, respectively. In the present study, we checked the convergent validity and discriminant validity for the twelve items. The results met the criteria (*Hair et al., 2009*), confirming the validity of all items (average variance extracted (AVE) > 0.5; square root of AVE > inter-construct correlations).

All items were scored on 7-point scales ranging from 1 (strongly disagree) to 7 (strongly agree).

### Behavior intention

We included another item regarding behavior intention toward the health proposal in the questionnaire ("In the future, when sanitizing my hands with alcohol-based hand sanitizer, I will press the pump slowly to the bottom to get a sufficient amount"). This item was also scored on a 7-point scale ranging from 1 (strongly disagree) to 7 (strongly agree).

## Materials

We selected the target health proposal message from a pilot study based on its relatively low knowledge rate (40.8%). Additionally, the attitude toward the message was rated as considerably favorable (i.e., the scores were significantly higher than 4 on a 7-point scale, $M = 5.35$, $SD = 1.29$, $t (200) = 5.52$, $p < 0.001$, Cohen's $dz = 1.047$), which qualified the message as a strong one and enabled us to further explore the effect of message repetition (*Cacioppo & Petty, 1989*; *Claypool et al., 2004*).

The content of the message in Japanese was: アルコールは消毒・殺菌効果があると言われています。一方、少量のアルコール消毒液を手に取った場合、消毒効果が低下してしまいます。そこで、アルコール消毒液で手を消毒する際は、少なくともポンプ部分を下までゆっくり押して、たっぷり手に取って使うことをおすすめします。 The English translation of the message is as follows: "While alcohol-based hand sanitizer is useful for preventing COVID-19 infection, the effect will be discounted if not enough amount is used. Therefore, it is recommended to press the pump to the bottom every time to get enough amount of hand sanitizer."

## Procedures

We conducted the two-wave surveys for the no-repetition and repetition conditions from October 22 to November 6, 2020. In the first wave, participants in two conditions answered the same dummy questionnaire of the Need for Cognition Scale. Only participants in the repetition condition were exposed to the target health proposal message at the end of the dummy questionnaire. After 24–72 h, participants in both conditions were exposed to the same message and required to fill out the questionnaire containing 13 items in Table 1. Additionally, two other questions were used as manipulation checks for message repetition. The first question was for all participants when they were exposed to the message for the first time: "Have you seen this message before?" The other question was asked at the end of the dummy questionnaire only in the first wave of the repetition condition: "What is the message about?"

As for the time interval between two waves (24–72 h in this study), we decided it based on the time point of participants' answers in the previous research. In a study that also discussed the effects of repeated fear-arousing message in the EPPM, the time interval was 3 days (*Shi & Smith, 2016*). However, the stimuli they used was a video clip. It is confirmed that pictures were remembered better than words in former studies (e.g., *Grady et al.,*

**Table 2 Correlations for SEM analyses.**

| Observed variable | 1 | 2 | 3 | 4 | 5 | 6 | 7 | 8 | 9 | 10 | 11 | 12 | 13 | 14 |
|---|---|---|---|---|---|---|---|---|---|---|---|---|---|---|
| 1. Item1 | 1 | | | | | | | | | | | | | |
| 2. Item 2 | 0.685 | 1 | | | | | | | | | | | | |
| 3. Item 3 | 0.754 | 0.863 | 1 | | | | | | | | | | | |
| 4. Item 4 | 0.415 | 0.397 | 0.443 | 1 | | | | | | | | | | |
| 5. Item 5 | 0.527 | 0.625 | 0.617 | 0.688 | 1 | | | | | | | | | |
| 6. Item 6 | 0.409 | 0.423 | 0.457 | 0.689 | 0.599 | 1 | | | | | | | | |
| 7. Item 7 | 0.308 | 0.281 | 0.305 | 0.143 | 0.151 | 0.202 | 1 | | | | | | | |
| 8. Item 8 | 0.262 | 0.187 | 0.243 | 0.076 | 0.113 | 0.114 | 0.668 | 1 | | | | | | |
| 9. Item 9 | 0.115 | 0.089 | 0.144 | 0.066 | −0.003 | 0.101 | 0.556 | 0.579 | 1 | | | | | |
| 10. Item 10 | 0.270 | 0.233 | 0.338 | 0.204 | 0.212 | 0.188 | 0.363 | 0.310 | 0.379 | 1 | | | | |
| 11. Item 11 | 0.357 | 0.380 | 0.441 | 0.240 | 0.348 | 0.283 | 0.310 | 0.312 | 0.267 | 0.708 | 1 | | | |
| 12. Item 12 | 0.304 | 0.285 | 0.371 | 0.204 | 0.233 | 0.220 | 0.362 | 0.304 | 0.368 | 0.847 | 0.731 | 1 | | |
| 13. Behavior intention | 0.552 | 0.599 | 0.710 | 0.482 | 0.506 | 0.462 | 0.227 | 0.122 | 0.158 | 0.373 | 0.378 | 0.436 | 1 | |
| 14. Repetition | 0.061 | −0.013 | 0.002 | 0.086 | 0.020 | 0.065 | −0.065 | −0.033 | −0.031 | −0.066 | −0.076 | −0.076 | −0.037 | 1 |

Note:
The variables were standardized to have a mean of 0 and a standard deviation of 1. $N = 326$.

**Table 3 Means and standard deviations for each item in SEM.**

| | Item 1 | Item 2 | Item 3 | Item 4 | Item 5 | Item 6 | Item 7 | Item 8 | Item 9 | Item 10 | Item 11 | Item 12 | Behavior intention |
|---|---|---|---|---|---|---|---|---|---|---|---|---|---|
| M | 5.74 | 5.53 | 5.65 | 4.61 | 5.20 | 4.71 | 4.82 | 5.13 | 3.96 | 5.56 | 5.99 | 5.66 | 5.50 |
| SD | 1.44 | 1.36 | 1.38 | 1.49 | 1.26 | 1.38 | 1.47 | 1.41 | 1.31 | 1.37 | 1.17 | 1.42 | 1.65 |

*1998*) and thus 3 days might be a little too long for a short and simple health proposal message to be well remembered. We set the time interval between two waves to 24 h, and the questionnaire in each wave was accessible in 24 h. Since the questionnaires were completed online, we could not control the precise time point at which participants answered the second questionnaire. Considering the possible time point that participants completed two questionnaires, the range of time interval for all participants was decided to be 24–72 h.

## RESULTS

Tables 2, 3 presents the correlations among variables with means and standard deviations. We chose a robust maximum likelihood estimator to perform SEM for the model in Fig. 1, which produced Satorra–Bentler rescaled $\chi^2$ to correct non-normality-induced bias. The model yielded an acceptable fit (*Kline, 2015*; *Bentler & Bonett, 1980*): $\chi^2$ (34, $N = 326$) = 196.367, $p < 0.001$; comparative fit index (CFI) = 0.949; Tucker–Lewis index (TLI) = 0.934, root mean square error of approximation (RMSEA) = 0.059 (90% CI [0.047–0.070], $p > 0.05$). The $p$ values of the coefficients of concern were adjusted using the

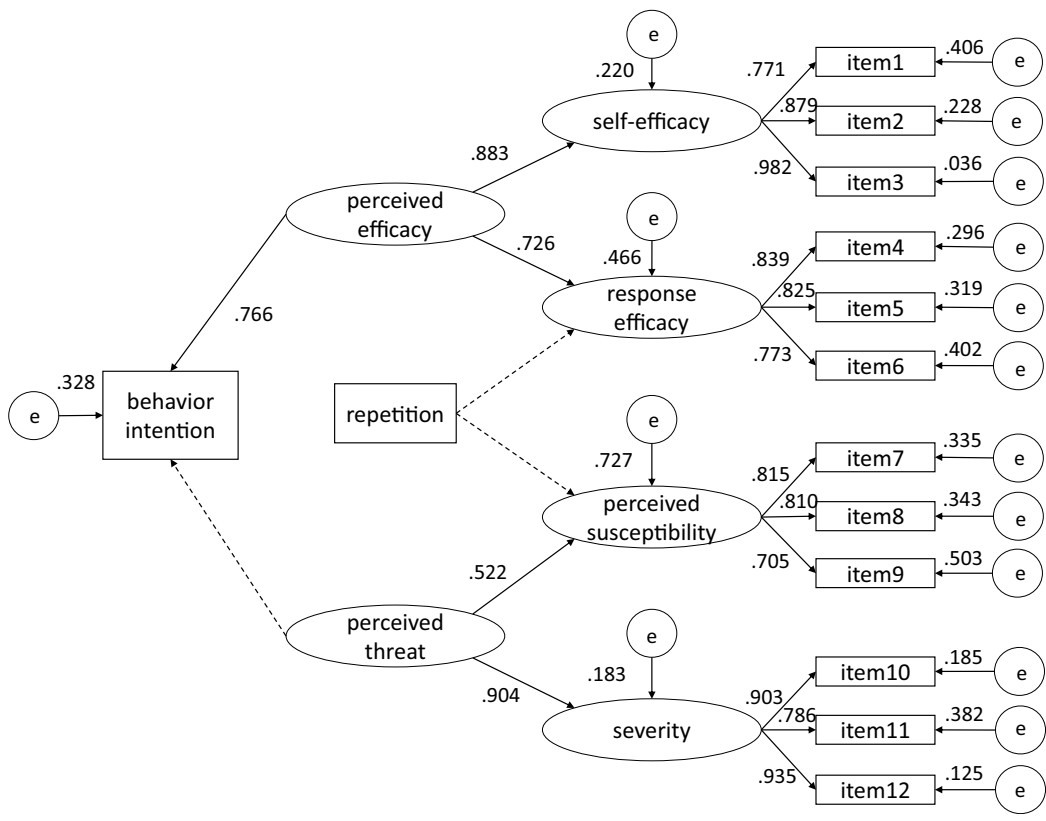

**Figure 2 The results from the theoretical model.** Standardized estimates of the model are listed above or to the left. Dotted lines indicate relationships that were not statistically significant.

false discovery rate (*Benjamini & Hochberg, 1995*). The results of the theoretical model are presented in Fig. 2.

It was shown that perceived efficacy had positive effects on both self-efficacy ($\beta = 0.883$, $p < 0.001$) and response efficacy ($\beta = 0.726$, $p < 0.001$). Similarly, perceived threat was found to positively influence perceived susceptibility ($\beta = 0.522$, $p < 0.001$) and severity ($\beta = 0.904$, $p < 0.001$). However, we did not detect a significant influence of message repetition on either response efficacy ($p = 0.110$) or perceived susceptibility ($p = 0.680$). Furthermore, the results confirmed the positive influence of perceived efficacy on behavior intention ($\beta = 0.766$, $p < 0.001$), but no effect of perceived threat on behavior intention ($p = 0.313$).

## DISCUSSION

In the present study, we performed SEM to examine whether the EPPM could be generalized to a naturalistic context like the COVID-19 pandemic. Furthermore, we explored how a repeated health proposal is involved with the EPPM. According to the results of our analyses, the sub-dimensions of perceived efficacy toward the target message and perceived threat toward COVID-19 were confirmed, which supported Hypotheses 1–4. There was no evidence revealing that message repetition significantly influenced

response efficacy (Hypothesis 5) or perceived susceptibility (Hypothesis 6). Furthermore, while we found a positive effect of perceived efficacy on behavior intention (Hypothesis 7), we did not detect a significant effect of perceived threat on behavior intention (Hypothesis 8).

Regarding the generalization of the EPPM in the COVID-19 context, most of our hypotheses were confirmed, except for the path from perceived threat to behavior intention. According to the EPPM, high threat and low efficacy would lead to a fear control process where people try to control their fear instead of taking efficient actions. In contrast, high threat and high efficacy would result in a danger control process where people take efficient actions to avert the threat. Moreover, low threat-high efficacy condition and low threat-low efficacy condition are not sufficient triggers for behavior change compared with the former two conditions. To the best of our knowledge, there is limited research using SEM to test the EPPM, except for a study conducted by *Ort & Fahr's (2018)*. Considering that the differences in analysis strategies may bring us new insights into the result of the non-significant path from perceived threat to behavior intention, we would take into account other analysis strategies in the previous research when discussing the results. Nevertheless, the following discussion was based on several post-hoc analyses, which were not registered in advance. We conducted the analyses simply for exploration, and thus any implication based on them should be treated with caution and confirmed through further investigation.

In several studies (*Gore & Bracken, 2005*; *Witte, 1996*), the average score of the six threat items was used as the score for perceived threat, and the average score of the six efficacy items was used as the score for perceived efficacy. Then the discriminating value was calculated by subtracting these two scores ($Z_{efficacy} - Z_{threat}$ = discriminating value). When the discriminating value was negative (i.e., perceived threat outweighed perceived efficacy), the participant was considered to get involved in the fear control process. When the discriminating value was positive (i.e., perceived efficacy outweighed perceived threat), the participant was considered to take danger control actions. In other words, the formula emphasized the comparison between perceived threat and perceived efficacy rather than a single variable's level (high/low) alone in determining health communication outcomes. Applying this strategy, we divided our participants in the no-repetition condition (which had the same design as the previous research) into two groups—danger control group and fear control group ($N_{danger}$ = 70, $N_{fear}$ = 82)—according to their discriminating values, and then conducted independent samples *t*-test on their behavior intention scores. The behavior intention of participants in the danger control group was significantly higher than that in the fear control group ($t(150) = 3.65$, $p < 0.001$), which was theoretically consistent with the basic frame of the EPPM. Given that these analyses were not registered beforehand and that the discriminating value approach has been criticized for its flawed conceptual logic (*Ort & Fahr, 2018*), we consider it not appropriate to make any conclusion about the EPPM's validity in the present study. Instead, we suggest treating the result's limited implication as a mere reference or a hint when discussing the relation between perceived threat and behavior intention.

There are also studies using the comparison between scores and the midpoint of the scale (four in the present research) to determine the level of perceived threat and perceived efficacy (*Shi & Smith, 2016*). When we applied this strategy, the results showed that only 16 out of 326 participants had low perceived efficacy toward the target health proposal message, and 3 out of 326 participants with low perceived threat toward COVID-19. The fact that few participants rated perceived threat low confirmed our expectation for the fixed properties of COVID-19—even with no extra fear-arousing material, its perceived threat stays at a high level. Moreover, in line with our pilot study's results that people have favorable attitudes toward the target message, the message's perceived efficacy was at a high level as well. In other words, we have mainly focused on the high efficacy-high threat condition in the present study.

The exploratory analyses mentioned above provide several possible explanations for the non-significant path between perceived threat and behavior intention in our theoretical model. As one possibility, a perceived threat may act as a judgment criterion, meaning that the danger control process will not start until the perceived threat reaches a certain level. Subsequently, the fluctuation in its value does not influence behavior intention anymore. If this is the case, the increase in perceived threat might not affect behavior intention, for the perceived threat was already at a high level in this study. Another possibility is that we have to take perceived threat's interrelation with perceived efficacy into consideration when evaluating its effect on behavior intention. When perceived threat increases and perceived efficacy is controlled to stay constant, there are three kinds of possible changes in the process people get involved in when facing a health threat, namely switching from the danger control process to the fear control process; staying unchanged in the fear control process; or staying unchanged in the danger control process. While behavior intention should decrease in the switch from the danger control process to the fear control process, it is unclear how behavior intention would change when the process stays the same. Consequently, the relation between perceived threat and behavior intention would not be monotonic. In any case, we prefer to think that threat has its own function even if not reflected in number because studies (*Witte, 1992*; *Gore & Bracken, 2005*) have shown that if individuals receive a no-threat and high-efficacy message, the message did not influence them, which is not predicted by the EPPM.

In all cases, we should not expect to enhance people's behavior intention through increasing COVID-19's perceived threat. Instead, it is more reliable to count on the effect of perceived efficacy on behavior intention toward effective health proposals, which is consistent with the results from previous research (*Earl & Albarracín, 2007*; *Mongeau, 1998*; *Peters, Ruiter & Kok, 2013*; *Ruiter et al., 2014*; *Tannenbaum et al., 2015*; *Ort & Fahr, 2018*).

Regarding message repetition, neither response efficacy nor perceived susceptibility were found to be influenced by message repetition. Based on the differences in message repetition manipulation between the present study and previous ones, there may be several possibilities accounting for the results. First of all, considering that there are 24–72 h between two exposures to the message, the stimulus was too simple to be impressive, making the manipulation of message repetition invalid. For example, in *Cacioppo &*

*Petty (1979)*, a simple message was repeated several times in one experiment section with almost no interval. In the studies where the interval lasted for several days or weeks (*Shi & Smith, 2016*; *Horowitz, 1969*; *Skilbeck, Tulips & Ley, 1977*), the materials serving as stimuli were more complicated (e.g., 5-min video clip; eight-page pamphlet; 20-min lecture). Similarly, in a recent study investigating the relation between short health reminders and the change in behavior intention during the COVID-19 pandemic (*Yonemitsu et al., 2020*), no effect of the reminders was reported with a 1-week interval. Therefore, it is reasonable to conjecture that in the present study, the complexity of the health proposal message and the length of time interval did not make a good match to test message repetition's effect. Furthermore, considering that COVID-19 has been a severe health threat in Japan—where the present study was conducted—the large amount of relevant information people received from sources other than the survey may have become an extraneous variable. On all accounts, it was not detected that mere repetition of a short piece of health proposal at 24–72h's interval significantly influenced its response efficacy and COVID-19's perceived susceptibility.

## CONCLUSIONS

### Practical implications

Before the present study, only one study (*Ort & Fahr, 2018*) had used SEM to show the inter-dependencies within the EPPM and to investigate the EPPM as a whole. This study confirmed the validity of this approach for data analysis in the field of health communication. Additionally, it is indicated that the EPPM can also provide precious insights into health compliance in the COVID-19 context and for the Japanese population.

The sub-dimensions of perceived efficacy were confirmed in the COVID-19 context. Combined with the effect of perceived efficacy on behavior intention, it is essential to improve both people's self-efficacy and proposal's response efficacy to achieve health compliance. Concretely, for a health proposal considered to be effective, "it can be easily carried out" and "it works well for preventing COVID-19 infection" are the points that need to be emphasized. Meanwhile, it was also confirmed that perceived threat consists of perceived susceptibility and severity. However, when the public's perceived threat for COVID-19 has already been at a high level, it may not help promote health compliance to show people how threatening it is or how easily they can get infected. Similarly, since we failed to detect the effect of message repetition in this study, it is necessary to take the efforts and resources spent on repeated messages with a grain of salt.

### Limitations and future studies

*Harper et al. (2020)* suggest that fear promotes public health compliance during the COVID-19 pandemic. In contrast to their results, we did not find perceived threat's effect on behavior intention toward the target health proposal in this study. While perceived threat may contribute to fear, the two concepts are not exactly the same. Therefore, future research should add fear in the model to confirm whether fear affects health compliance.

Rather than actual behavior change, the primary dependent variable in the model was behavior intention toward the target proposal. Since it is pointed out that there are cases where behavior intention was not a good approximation to actual behavior (*Leventhal, Singer & Jones, 1965*; *Evans et al., 1970*), future studies should try to include actual behavior index in the model. Moreover, only one health proposal was used as stimulus material in this study. We suggest that the effect of various health proposals should be addressed in subsequent studies. Specifically, considering that there was no association between message repetition and the EPPM in this study, one important task will be to explore the balance among the message's complexity, the length of time interval, and even the times of exposure to confirm the results.

In the exploration of public's irrational panic and response to major emergencies, a phenomenon called the "psychological typhoon eye" effect has been reported (*Li et al., 2009*). This phenomenon is defined as a tendency that residents living closer to the center of the devastated areas show less concern about safety and health. Former researchers have detected the psychological typhoon eye effect in the COVID-19 pandemic in China, showing a negative correlation between exposure level and mental health problems (*Wang et al., 2020*; *Zhang et al., 2020*). Moreover, it is indicated that the differences among residents in various areas were only found in the indicator, "wanting to have a physical investigation," but not in other behavioral response indicators (e.g., frequently washing or cleaning hands; taking the initiative to avoid strangers). Since the present study did not collect information about participants' residence, we could not compare the results of participants with various exposure levels to COVID-19. To better understand the mechanism of responses toward major emergencies like the COVID-19 pandemic, and as a result, promote public compliance with effective health proposals, we suggest future studies consider participants' residence when discussing the interrelation among variables.

### Funding
This research is supported by JSPS KAKENHI: JP19K14482 to Kyoshiro Sasaki, and JP16H03079, JP17H00875, JP18K12015, and JP20H04581 to Yuki Yamada. The funders had no role in study design, data collection and analysis, decision to publish, or preparation of the manuscript.

### Grant Disclosures
The following grant information was disclosed by the authors:
Kyoshiro Sasaki: JSPS KAKENHI, JP19K14482.
Yuki Yamada: JP16H03079, JP17H00875, JP18K12015 and JP20H04581.

### Competing Interests
Yamada Yuki is an Academic Editor for PeerJ. The authors declare that they have no competing interests.

## Author Contributions

- Jingwen Yang conceived and designed the experiments, performed the experiments, analyzed the data, prepared figures and/or tables, authored or reviewed drafts of the paper, and approved the final draft.
- Xue Wu conceived and designed the experiments, performed the experiments, analyzed the data, prepared figures and/or tables, authored or reviewed drafts of the paper, and approved the final draft.
- Kyoshiro Sasaki conceived and designed the experiments, performed the experiments, analyzed the data, prepared figures and/or tables, authored or reviewed drafts of the paper, and approved the final draft.
- Yuki Yamada conceived and designed the experiments, performed the experiments, analyzed the data, prepared figures and/or tables, authored or reviewed drafts of the paper, and approved the final draft.

## Human Ethics

The following information was supplied relating to ethical approvals (i.e., approving body and any reference numbers):

The present study received approval from the psychological research ethics committee of the Faculty of Human-Environment Studies at Kyushu University, Fukuoka, Kyushu, Japan (approval number: 2019-034).

## Data Availability

The raw measurements are available in the Supplemental File.

## Supplemental Information

Supplemental information for this article can be found online at http://dx.doi.org/10.7717/peerj.11559#supplemental-information.

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
