# Peer review of "No significant association of repeated messages with changes in health compliance in the COVID-19 pandemic: a registered report on the extended parallel process model"

_PeerJ, doi:10.7717/peerj.11559_

## Round 0.1 · original submission · Major Revisions

Thank you for your submission. The reviewers have presented a number of recommendations / questions for you to address in your re-submitted article.

Reviewer 1 ·

Basic reporting

No comment

Experimental design

No comment.

Validity of the findings

No comment.

Additional comments

Yang et al. examined the effects of repetition on behavior intent to adhere to a health message, in the context of perceived threat and efficacy as operationalized in the EPP model. The authors performed SEM to extricate interactions between different variables on behavioral intent. Interestingly, they find no effect of either repetition or threat on behavioral intention. These null findings, in the context of the Covid19 pandemic, are important and worthy of highlighting. Why waste time and resources on repetition or, worse, inducing a negative state of mind (fear), if these efforts have no effect? My main criticism here is that these null findings are not highlighted enough.

Some minor issues:
1. Given there was a null finding concerning the repetition of the health message I recommend changing the title of the study.

2. Some confusion about lines 345-351. I think it is interesting to speculate on causes, but it may also be that threat is simply not an effective motivator relative to self or response efficacy. There must be considerable literature in this area.

3. Also, while you mention Tannenbaum et al.’s meta-analysis, it seems you omit their supporting result that repetition of fear message has no effect. https://psycnet.apa.org/doiLanding?doi=10.1037%2Fa0039729

4. In line 414, the tendency towards Typhoon Eye is a bit speculative as well, as those most effected by a tragedy may leave maladaptive responses –such as panic- for a later time, when the immediate danger subsides.

5. Finally, the manuscript is well written but I recommend a very thorough copy-edit to streamline the wording.

·

Basic reporting

a. The authors tend to use “it” and “they” in reference to psychological constructs (like perceived threat on line 120). The authors should always label constructs instead of using pronouns so as to reduce confusion and aid reader comprehension.
b. It is not clear what “(1.47%)” on line 24 is in reference to. I believe that the authors are trying to state the Japanese mortality rate, but I am not sure. It would be worthwhile for the authors to label this number more clearly for the readers.
c. Lines 25-27 contain a confusingly worded sentence that ought to be revised, especially the phrase “achieved staged success.”
d. The authors should include a citation in line 31 after they define “functional fear.”
e. I cannot determine if line 38 should be revised so that it says “fear-appealing communication” or if “fear appealing communication” is the standard term from previous literature.
f. Lines 43 – 46 report about the positive and the negative main effects of fear, but it is not clear what the terms “positive” and “negative” are in reference to. I felt like “positive main effect of fear on persuasion” may mean that fear increases persuasiveness, but positive and negative are too ambiguous of terms to know for sure. The authors should specifically report the results from the studies they’ve cited in lines 43 – 46 in order to aid reader comprehension.
g. Line 47 contains the term “fear appeal,” but it seems to me that this should be revised to “fear-appealing communication.”
h. Line 328 should be revised to say “When we applied this strategy the results showed . . .”
i. The authors do a good job at highlighting the unique and specific contributions of their research.
j. The authors do a good job of explaining the EPPM in the paragraph contained within lines 50 – 65.
k. It is not clear if the authors mean to say that we are exposed to health proposals about COVID-19 multiple times per day or that people have been exposed to health proposals about COVID-19 at least once (lines 67-69).
l. It is very helpful that the authors reported all results in reference to their stated hypotheses then discussed those results more specifically.
m. I appreciated that the authors explicitly warned the reader about post-hoc exploratory analyses before discussing the results of those analyses (lines 295-302).
n. The paragraphs between lines 337 – 357 provide an excellent set of explanations for the non-significant path between perceived threat and behavior intentions.

Experimental design

a. The authors pose their hypotheses in lines 87 – 114 but then pose further hypotheses and/or research questions in lines 116 – 148. The authors may want to re-order the final part of their introduction so as to end with their hypotheses or label the final three paragraphs according to the research questions they include. At the moment, it feels as though the authors are posing additional research directions beyond the hypotheses they’ve already stated and I feel distracted from the important research hypotheses as I go into the next section.
b. Lines 125 – 127 may be re-stating a hypothesis or may be posing a research question. The authors should either reference their hypothesis directly or explicitly pose a new research question.
c. The authors do not explain why they chose to use the 24-72 hour delay between message exposures in this experiment. In light of their own discussion of previous research using different procedures (lines 364 – 373), it is necessary for the authors to explicitly defend their decision to use a 24-72 hour delay between message exposure for participants in this study instead of using previously successful procedures.
d. The authors’ arguments for why they removed some participants prior to analyses are clear and sufficient.
e. The authors do not report Cronbach’s alpha for their Risk Behavior Diagnosis Scale, but their reported analyses for convergent and discriminant validities are sufficient.
f. The authors’ explanation for why they chose their particular message is clear and sufficient.
g. The authors’ explanation for why they used their particular procedures is clear and sufficient.
h. The authors report that Table 2 contains correlations among variables along with means and standard deviations, but values for means and standard deviations appear to be missing in Table 2. There is only one value per cell and I believe that these values are the correlations. The authors should make certain to add the means and standard deviations for each item or report them in a separate table.

Validity of the findings

a. The authors do a good job of highlighting the important and relevant implications of their research.
b. The authors suggest useful directions for future studies.

Additional comments

The authors have done a good job and I believe that this paper will be suitable after minor revisions.

·

Basic reporting

OK, see General Comments

Experimental design

OK, see General Comments

Validity of the findings

Some issues, see General Comments

Additional comments

This is an interesting study, applying the EPPM in combination with a repetition of the behavior change message. Using the EPPM as well as introducing repetition within this framework are both commendable. I do have some issues with how the authors interpreted the EPPM and with their analyses. Those issues may be repairable.

The EPPM is a model predicting the effect of risk information on behavior, distinguishing between four predictors, combined in two predictors of intention. The way the authors present these four predictors in the hypotheses, as all having an independent influence on intention, is misleading. The essence of the all predictions of the EPPM lies in the interaction of those four concepts. For example, even if severity, susceptibility, response efficacy are positive and high, there would not be a behavior change when self-efficacy is low, et cetera. I think that it would help the reader when the authors or more explicit about the crucial role of the interaction among the predictors.

Related to that first issue, I cannot get a clear idea how the authors deal with these interactions in the SEM analysis. E.g. is perceived efficacy an outcome of the SEM or just the combination of those six questions?

The results are a nice demonstration of what the EPPM would predict: people don’t change as a result of threat or fear, but as a result of efficacy, especially self-efficacy, given that the threat is there anyway. The total lack of effect from repetition is obviously unexpected, but my interpretation is that the manipulation of repetition is rather weak.

---

## Round 0.2 · accepted · Accept

Thank you for your revised submission. The reviewers are happy with the changes and therefore I am happy to accept your paper for publication.

Reviewer 1 ·

Basic reporting

No comment

Experimental design

No comment

Validity of the findings

No comment

Additional comments

No comment

·

Basic reporting

The authors' revisions have adequately addressed all my concerns.

Experimental design

The authors' revisions have adequately satisfied all of my concerns.

Validity of the findings

The authors' revisions have adequately satisfied all of my concerns.

Additional comments

Thank you for your thoughtful and thorough revisions.

·

Basic reporting

See general comments.

Experimental design

See general comments.

Validity of the findings

See general comments.

Additional comments

The authors have responded to my comments by repeating their earlier interpretation. That is fine with me, given that they explain carefully what they do and why.
There is one point that seems incorrect: “In any case, we prefer to think that threat has its own function even if not reflected in number because studies (Witte, 1992; Gore & Bracken, 2005) have shown that if individuals receive a no-threat and high-efficacy message, the message did not influence them, which is not predicted by the EPPM (380-382)”. I do not think that last remark is correct. There is more in life than threat, and if there were no other reason for behavior than no-threat, why would people do something? High self-efficacy alone is no reason for behavior.
We need research papers on COVID-19 preventive behaviors and we need them fast. Readers can check for themselves if they agree with the authors’ interpretation.